# Combined Humeral Head and Shaft Fractures: Outcome Following Intramedullary Nailing and Plating

**DOI:** 10.3390/medicina59010113

**Published:** 2023-01-05

**Authors:** Firas Souleiman, Jan Theopold, Ralf Henkelmann, Georg Osterhoff, Torsten Pastor, Boyko Gueorguiev, Johannes Fakler, Pierre Hepp

**Affiliations:** 1Department of Orthopaedics, Trauma and Plastic Surgery, University Hospital Leipzig, 04103 Leipzig, Germany; 2AO Research Institute Davos, 7270 Davos, Switzerland; 3Department of Orthopedic and Trauma Surgery, Lucerne Cantonal Hospital, 6000 Lucerne, Switzerland

**Keywords:** proximal humerus, intermuscular humerus fracture, outcome, intramedullary nail, plate fixation, complications

## Abstract

*Background and Objectives*: Combined fractures of the humeral head and shaft (FHS) are rare but frequently involve an intermuscular fracture as its characteristic pattern. The aim of this retrospective study was to investigate intramedullary nailed and plated FHS in terms of outcomes and complications. *Materials and Methods*: The present study included patients with FHS, treated via either intramedullary nailing or plating within a period of 10 years, with a minimum follow-up of 12 months. Functional outcome was assessed using the age- and sex-adapted Constant–Murley Score (CMS-K). Rates of complications and revision surgeries were registered. *Results*: Twenty-five patients (18 females, 7 males, age 60.1 ± 14.2 years, range 23–76 years) were included in the study. Nailing was performed in 16 patients (12 females, 4 males, age 62.6 ± 12.4 years), whereas plating was executed in nine patients (6 females, 3 males, age 55.8 ± 17.0 years). Follow-up among all patients was 45.1 ± 26.3 months (range 12–97 months). CMS-K was 70.3 ± 32.3 in the nailing group, with reoperation in four cases, and 76.0 ± 31.0 in the plating group, with one reoperation (*p* = 0.42). Patients with no metaphyseal fragment displacement (*n* = 19; CMS-K 76.7 ± 17.3) demonstrated significantly better functional outcomes versus those with secondary displacement of the metaphyseal fragments (*n* = 6; CMS-K 60.0 ± 17.1), *p* = 0.046. *Conclusions*: Comparable acceptable clinical outcome is obtained when comparing nailing with additional open cerclage or lag-screw fixation techniques versus plating with open reduction. However, a higher revision rate was observed after nailing. The correct metaphyseal fragment fixation seems to be crucial to avoid loss of reduction and hence the need for revision surgery, as well as a worse outcome.

## 1. Introduction

Proximal humerus fractures constitute approximately 4–5% of all fractures and frequently indicate the presence of osteoporosis [1]. In addition to the increasing incidence of these fractures, more complex morphologies are associated with the osteoporotic bone structure [2,3,4]. Several studies have described combined fractures of the humeral head and shaft (FHS) [5,6]. Their frequent pattern involves the so-called intermuscular fracture, hypothesized to result from a sudden pull of the pectoralis and rotator cuff muscles [6,7]. Conservative therapy is preferred for non-dislocated fractures, whereas surgical treatment may be performed in dislocated fractures [8]. Treatment with intra- and extramedullary implants has been investigated in previous reports, predominantly focusing on isolated proximal humerus or shaft fractures [7,9,10,11]. Combined proximal and shaft fractures—involving intermuscular fractures—have rarely been described and related to inconsistent classification systems and treatment algorithms [5,6,7].

Intermuscular fractures of the metaphyseal humerus are characterized by diverging fragments of the metaphyseal region due to traction by the muscles surrounding the shoulder. Fracture morphologies vary depending on the extent of the pull of individual shoulder muscles (Figure 1 and Figure 2). Usually, anterior fragments are displaced by the pectoral muscles and the posterior fragments—by the deltoid and latissimus dorsi muscles. The forces exerted by these muscles cause a higher degree of displacement and can make exact anatomical reconstruction more challenging and prone to complications [7].

Commonly, fixation of these fractures is achieved via intramedullary nailing or plating, and—depending on the extent of displacement—additional cerclages or lag-screws are placed [5,6,7].

The aim of this retrospective study was to investigate intramedullary nailed and plated FHS in terms of outcomes and complications of the intermuscular fractures.

## 2. Materials and Methods

### 2.1. Patients

This retrospective study was conducted at a primary level trauma center with approval from the local ethics committee (494/16-ek) and in accordance with the principles of the Declaration of Helsinki. All consecutive patients who had been surgically treated for FHS were identified from the institutional databases. Informed consent was obtained from all individual participants included in the study. Inclusion criteria were age above 18 years, fixation of FHS performed between January 2004 and December 2013, normal shoulder function prior to the trauma, and a minimum follow-up of 12 months after surgery. Exclusion criteria were isolated humeral head or shaft fracture, fragment displacement of less than 5 mm, or an axis tilt of less than 20°.

### 2.2. Baseline Data Acquisition

Age, sex, accident mechanism, concomitant injuries, American Society of Anaesthesiologists (ASA) score, and comorbidities were obtained by chart review. Fracture morphologies were classified according to the Neer [12], AO/OTA [13], Garnavos/Lasanianos [6], and Stedtfeld [7] classifications by two experienced orthopaedic trauma surgeons. Patients were assigned to two groups corresponding to intramedullary nail (NO) and plate (PO) osteosynthesis.

### 2.3. Surgical Procedure

After positioning the patient in beach-chair position, a deltasplit approach with closed fracture reduction was performed for nailing, while a deltoideopectoral approach with open fracture reduction was performed for plating. Targon PH (Aesculap, Tuttlingen, Germany) and Retrograde Expert Nail (DePuy Synhtes, Zuchwil, Switzerland) were used for nailing. Winsta PH (Axomed, Freiburg, Germany) and in one case PHILOS (DePuy Synthes, Zuchwil, Switzerland) were implanted for plating. If the fracture was identified intraoperatively as highly vulnerable to intermuscular fragment dislocation, additive fixations were applied to the main implant (nail or plate). Cerclages, metaphyseal screws, or additive plates were used for this purpose. To antagonize the muscular tractions, additive suture fixation of the rotator cuff was performed during plate osteosynthesis.

### 2.4. Outcome

Rates of postoperative complications and reoperations were registered by chart review. At follow-up visits, shoulder function of all patients was assessed using the Constant–Murley Score, being age- and sex-adapted according to Katolik (CMS-K) [14]. On standard anterior–posterior and axial radiographs, implant failure and fragment displacement were assessed (Figure 3 and Figure 4). Relevant secondary displacement was defined as a fragment displacement of more than 5 mm or an axis tilt of more than 20° [15].

### 2.5. Statistical Analysis

Statistical analysis was performed with SPSS software package (V.24, IBM, NY, USA) used for calculations and graphical presentations of the results. The two groups of patients were characterized by descriptive statistics in terms of mean value ± standard deviation (SD) for continuous variables and rate (%) for categorical variables. Mann–Whitney test was applied for detection of significant differences between the groups. A subgroup analysis was performed for cases with additional metaphyseal fixation (cerclage, screws, plates) to evaluate whether correct metaphyseal fixation led to better functional results. Level of significance was set to 0.05 for all statistical tests.

## 3. Results

In total, 45 patients with FHS were identified in the databases between 2004 and 2013. Twenty-five patients (18 females and 7 males, age 60.1 ± 14.2 years, range 23–76 years) with an adequate follow-up were included in the final analysis. The reasons for the drop-out of patients are visualized in Figure 5.

High-energy trauma was the cause of fracture in nine patients (36.0%). ASA score among all patients was 2.2 ± 0.6. The distribution of fracture patterns according to the Neer, AO/OTA, Garnavos/Lasanianos, and Stedtfeld classifications is summarized in Table 1.

Intramedullary nailing was performed in 16 cases, whereas plate fixation was used in 9 cases.

No significant differences were detected between the groups in terms of age, sex, ASA score, trauma mechanism, and follow-up period (Table 2).

The overall CMS-K of both groups was 72.6 ± 18.4 at a follow-up of 45.1 ± 26.3 months (range, 12–97 months). The CMS-K was 70.3 ± 32.3 in the NO group and 76.0 ± 31.0 in the PO group (*p* = 0.42) with a comparable follow-up time between them (*p* = 0.98). Combined fractures according to Garnavos et al. were treated with intramedullary nailing or plating in three cases each. CMS-K was 83.3 ± 13.5 in case of NO and 71.3 ± 25.1 in case of PO. Extended fractures according to Garnavos et al. were treated in 13 cases with a nail fixation (CMS-K, 67.2 ± 16.0) and in six cases with plate osteosynthesis (CMS-K 79.8 ± 21.9, *p* = 0.24). Additive fixation was used in 17 cases, including cerclages (NO: 3/16, PO: 2/9), metaphyseal lag-screw (NO: 6/16, PO: 4/9), and double-plate fixation (only PO: 2/9).

CMS-K was 76.7 ± 17.3 for patients without secondary metaphyseal displacement during the follow-up (*n* = 19). In cases of metaphyseal displacement and insufficient fracture healing (*n* = 6), the CMS-K was significantly lower (60.0 ± 17.1, *p* = 0.046).

Reoperation was necessary for five cases (5/16) in the NO group and in one case (1/9) in the PO group. The reasons for revision of nail osteosynthesis were displacement of metaphyseal fragments due to lack of fixation with change to either plate osteosynthesis (*n* = 2), inverse prothesis (*n* = 1), or removal of the implants (*n* = 1). In one case, the nail displaced proximally after loss of reduction. The reason for one revision in the PO group was nonunion due to metaphyseal lack of fixation. All complications in both groups occurred after a large extension of the fracture split into the shaft (extended C-fractures in the shaft region, *n* = 6).

## 4. Discussion

This study aimed to evaluate the surgical treatment of nailed and plated combined FHS. Both fixation techniques resulted in a good outcome when anatomical metaphyseal fixation has been achieved. Depending on the exact morphology, each technique seems to have certain advantages. Further, a subgroup analysis was performed to detect whether an anatomical reconstruction of a typical metaphyseal fracture component influences the outcome. In cases with secondary metaphyseal displacement, the functional outcome was significantly worse regardless of the used osteosynthesis hardware (nail or plate).

The follow-up rate of 56.0% is lower compared to other studies with a range of 86.7–100.0% [6,11]. Reasons for this are that those studies did not implement clinical outcome scores and had shorter follow-up periods. While in the referenced work the follow-up ranged between 3 and 17 months, it was up to 97 months in the presented study. The longer follow-up period is associated with a high mortality rate of 24.4% (*n* = 11), suggesting a complicated, multimorbid patient population [16]. Epidemiological studies on proximal humerus fractures confirmed unexpectedly high mortality rates of up to 9.5% in the first postoperative year [16]. Increasing numbers of multimorbid patients with more than three secondary diseases in up to 22.9% are described [17].

In the literature, high-energy trauma rates of approximately 25% are reported for proximal humerus fractures, increasing with the complexity of the fracture pattern [17]. This is in line with the high percentage of FHS caused by high-energy trauma in this and other studies [17,18].

The fracture morphologies in the presented work demonstrated high heterogeneity, making any classification difficult. Recent classification systems attempt to describe the FHS [6,7]. Garnavos et al. made the distinction between “extended” and “combined” fractures in order to describe the relationship between head and shaft fractures [6]. Their system specifies whether metaphyseal involvement is present or not; however, it does not describe the detailed morphology. “Extended” fractures are multi-fragmentary fractures with an intermuscular component and increased risk of displacement. Compared to the work of Garnavos et al., slightly fewer combined injuries occurred in the current study [6].

In contrast, Stedtfeld et al. classified 17 morphological subgroups based on the AO/OTA classification [7]. Their study included 10% A, 44% B1, 10% B2, 12% C, and 24% D fractures [7]. Our study included more complex C (24%) and D (44%) fractures.

Consideration of fracture morphologies caused by shoulder muscles seems to have an important influence on the most adequate fixation and is currently not represented in any classification system [7].

Studies examining outcome after surgical treatment of FHS are rare [6,11]. Constant–Murley scores of 71.2 (*n* = 72) and 74.4 (*n* = 18), reported after intramedullary nailing at least one year postoperatively, are comparable with the current study [6]. A previous investigation examining the outcome following plating is difficult to relate to our study because shoulder-specific functional outcomes were not assessed there [11].

In the present work, both fixation methods resulted in a good functional outcome in terms of age- and gender-adapted Constant–Murley Score by using additional metaphyseal fixation. General comparative studies between nail and plate osteosynthesis of proximal humeral fractures have demonstrated slight advantages in favor of the plate osteosynthesis with increasing fracture complexity [18,19,20].

The morphological subdivision according to Garnavos et al. demonstrated mild advantages in favor of the nail osteosynthesis of FHS in the current study. Regarding the treatment of extended fractures, some advantages in favor of the plate osteosynthesis were identified.

Extended fractures with a C-component of the metaphyseal shaft according to the AO/OTA classification demonstrated higher rates of displacement and revision after treatment with nail osteosynthesis due to the high torsional forces of the deltoideus and pectoralis major prevailing in this region [7]. Plating may therefore neutralize these forces and moments due to the various options for proximal and metaphyseal screw placement, as well as the ability to provide external splinting [11,21]. The results reveal that a secondary displacement or malunion in the metaphyseal region has a significant negative influence on the outcome (CMS-K score) of FHS osteosynthesis. In six cases (24%), displacement was observed postoperatively—independently of the used implants. All of these fractures were “extended” according to Garnavos et al. [6]. This supports the thesis that during preoperative analysis special attention must be paid to the displacement caused by the corresponding muscles. Correct metaphyseal fixation is sometimes difficult, as many muscles in the region of the humeral metaphysis exert forces and have to be antagonized surgically [7]. Additive screws, cerclages, and plates are possible at the expense of larger surgical approaches—as used in this investigation.

The main limitation of this study is related to its retrospective monocentric design. The analysis of complications was based on the data available in the medical records with no possibility of prospective observation. The study period was about a decade, finished almost 10 years ago. Nowadays, more advanced implants are available for treatment of complex proximal humerus fractures. Especially, cement augmentation and bone grafting of osteoporotic fractures as well as advanced implant designs enhancing better stability were developed [22,23,24,25]. It would be interesting to investigate patients with FHS after fracture stabilization with such implants in future studies. The follow-up period had a wide range from 12 to 97 months, which could allow varying outcome scores due to different time points of the follow-up. However, the average follow-up time between the groups was comparable. Due to the rare fracture morphology, the number of included patients was small but still comparable to studies published so far [5,6,7]. The main reason for the small number of patients was the high number of lost ones for follow-up due to inadequate preoperative shoulder function, death, or non-domestic patients with missing contact details. The use of different implants with different screw positions and different metaphyseal fixation options further increased the heterogeneity of the investigated cohort.

## 5. Conclusions

The combination of humeral head and shaft fracture presents a special challenge for orthopaedic trauma surgeons. This study demonstrates that the classic closed reduction and intramedullary nailing is not the best treatment option. Comparable acceptable clinical outcome was obtained when comparing nailing with additional open cerclage or lag-screw fixation techniques versus plating with open reduction. However, a higher revision rate was observed after nailing. The correct metaphyseal fragment fixation seems to be crucial to avoid loss of reduction and hence the need for revision surgery, as well as a worse outcome.

## Figures and Tables

**Figure 1 medicina-59-00113-f001:**
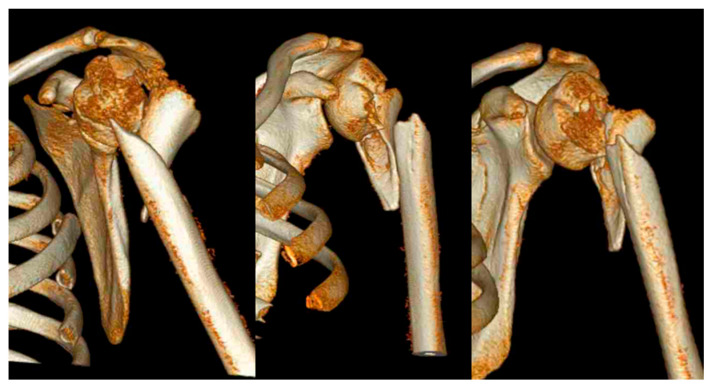
Example of a 3D computed tomography reconstruction of a combined intermuscular humerus fracture according to Garnavos et al. [6]. Shown is the typical intact bony segment between the fractured humeral head and the fractured diaphysis.

**Figure 2 medicina-59-00113-f002:**
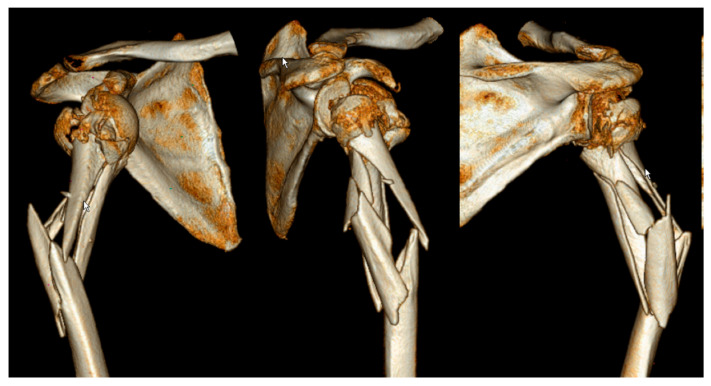
Example of a 3D computed tomography reconstruction of an extended dislocated intermuscular humerus fracture according to Garnavos et al. [6]. Shown is the typical fracture extension from the fractured humeral head to the middle third of the humeral diaphysis. Divergent fragment displacement in the region of the proximal metaphysis is typical for extended intermuscular fractures due to the attached muscles.

**Figure 3 medicina-59-00113-f003:**
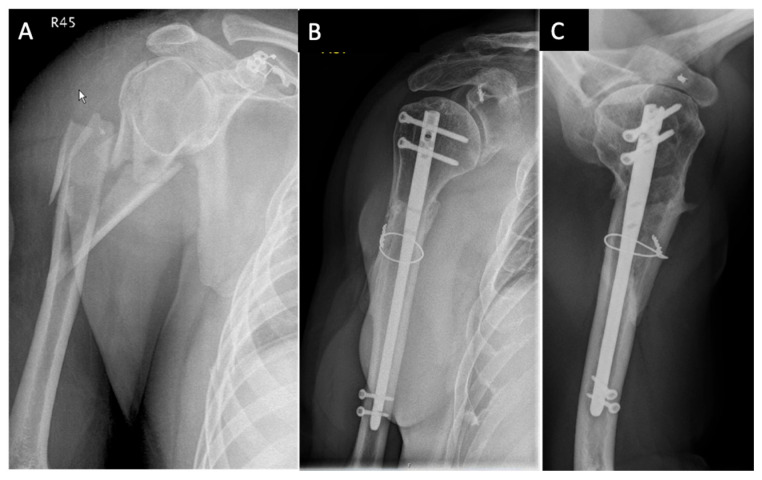
Example of surgical treatment with nailing and a metaphyseal cerclage: (**A**) Anterior–posterior X-ray image of an extended D3 fracture according to the Stedtfeld classification of a 43-year-old man after high energy trauma. (**B**) Anterior–posterior X-ray image 1 year after treatment with nail osteosynthesis and metaphyseal stabilization by a cerclage. (**C**) Axial X-ray image one year postoperative.

**Figure 4 medicina-59-00113-f004:**
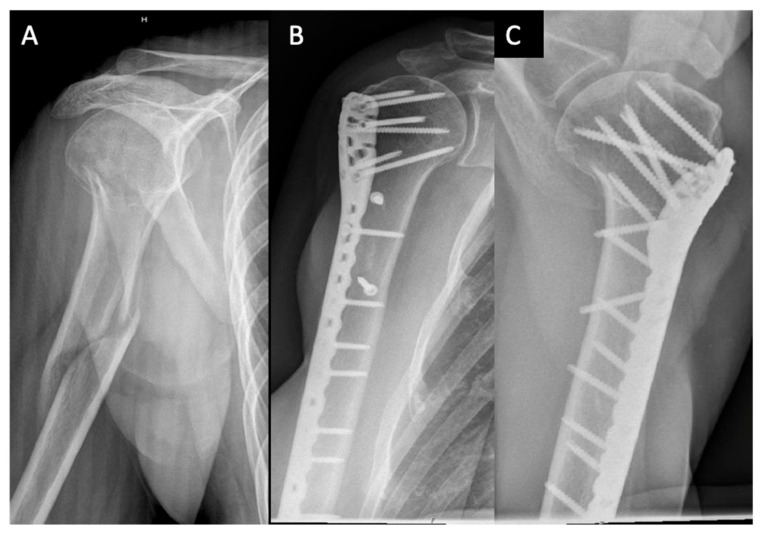
Example of surgical treatment with plating: (**A**) Anterior–posterior X-ray image of an extended D2 fracture of a 50-year-old woman according to the Stedtfeld classification. (**B**) Anterior–posterior X-ray image 16 months postoperative after treatment by plate osteosynthesis and additive metaphyseal screw stabilization. (**C**) Axial X-ray image 16 months postoperative after treatment by plate osteosynthesis and additive metaphyseal screw stabilization.

**Figure 5 medicina-59-00113-f005:**
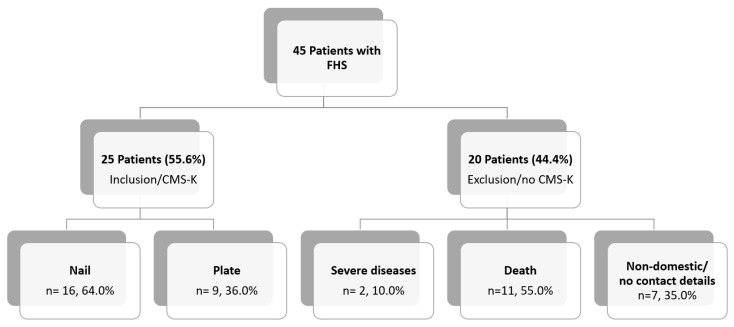
Reasons of inclusion/exclusion of patients with FHS from 2004 to 2013. FHS: combined fractures of humeral head and shaft; CMS-K: Constant–Murley Score, age- and gender-adapted according to Katolik.

**Table 1 medicina-59-00113-t001:** Patient characteristics; fracture classifications according to Neer, AO/OTA proximal and metaphyseal, Stedtfeld, and Garnavos/Lasanianos; used implants; follow-up with outcome; and observed revisions.

Case	Age, years	Sex	Neer	AO/OTA Proximal	AO/OTA Metaphyseal	Stedtfeld	Garnavos/ Lasanianos	Implant/ Metaphyseal Fixation	FU, Months/ CMS-K	Revision
1	65	M	IV	11-B1	12-C2	D4	Combined	Retrograde nail	48/83	-
2	74	F	III	11-A3	12-C3	A2	Extended	T-nail	97/51	Metaphyseal lack of fixation, material removal
3	68	F	IV	11-B1	12-C1	D2	Extended	T-nail	32/71	Metaphyseal lack of fixation, change to plate
4	43	M	IV	11-B1	12-C3	D3	Extended	T-nail/cerclage	92/77	-
5	74	F	V	11-B2	12-C1	B1-3	Extended	T-nail/screw	52/64	-
6	65	M	III	11-A2	12-A1	A2	Extended	T-nail/screw	60/62	-
7	75	F	III	11-A2	12-C1	C3	Extended	T-nail	91/64	-
8	72	F	VI	11-C3	12-C3	D3	Extended	T-nail/screw	55/68	Metaphyseal lack of fixation, inverse prothesis
9	55	F	III	11-A2	12-C1	B2-1	Extended	T-nail	70/99	-
10	67	F	III	11-A2	12-C1	C2	Extended	Locking plate/ cerclage, lag screw	63/99	-
11	37	F	IV	11-B1	12-C1	D3	Extended	T-nail/screw	34/86	-
12	50	F	III	11-A2	12-C1	C4	Combined	T-nail/cerclage	49/97	-
13	76	F	V	11-C2	12-C2	D4	Combined	Locking plate/ lag screw	58/65	-
14	71	M	IV	11-B1	12-C1	D4	Extended	T-nail/screw	15/38	Loss of reduction, proximal nail displacement, material removal
15	70	F	IV	11-B1	12-C2	D4	Combined	T-nail/cerclage	15/70	-
16	23	M	IV	11-B2	12-C2	D4	Combined	Locking plate/ plate	35/99	-
17	46	F	V	11-C2	12-C3	B1-3	Extended	Locking plate/ lag screw	57/100	-
18	76	F	III	11-A2	12-C1	C2	Extended	T-nail	12/72	-
19	53	F	III	11-A3	12-C3	C3	Extended	T-nail/screw	12/48	Metaphyseal lack of fixation, change to plate
20	53	F	IV	11-B1	12-A1	B2-3	Extended	T-nail	12/74	-
21	51	F	IV	11-B1	12-C1	D2	Extended	Locking plate/ lag screw	37/89	-
22	49	M	V	11-C2	12-C3	D2	Combined	Locking plate/ plate	61/50	-
23	48	F	V	11-C2	12-C3	B1-3	Extended	Locking plate	45/84	-
24	68	M	IV	11-B1	12-C1	B1-2	Extended	Locking plate/ lag screw	12/45	Metaphyseal lack of fixation, pseudarthrosis
25	74	F	III	11-C1	12-C3	C3	Extended	Locking plate/ cerclage	13/62	-

FU—follow-up; CMS-K—age- and sex-adapted Constant–Murley Score.

**Table 2 medicina-59-00113-t002:** Patient characteristics in the NO (Nail) and PO (Plate) groups together with the corresponding *p*-values.

Characteristics	Nail (*n* = 16)	Plate (*n* = 9)	*p*-Value
Age, years	62.6 ± 12.4	55.8 ± 17.0	0.33
Sex	12F, 4M	6F, 3M	0.76
High-energy trauma	*n* = 6, 37.5%	*n* = 3, 33.3%	–
ASA	2.1 ± 0.7	2.2 ± 0.4	0.85
Follow-up, months	46.6 ± 29.9	42.3 ± 19.7	0.98
CMS-K	70.3 ± 32.3	76.0 ± 31.0	0.42
Postoperative revision	*n* = 5; 31.3%	*n* = 1; 11.1%	–

## Data Availability

Not applicable.

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
