# Peer review of "Combined Humeral Head and Shaft Fractures: Outcome Following Intramedullary Nailing and Plating"

_medicina, 2023, doi:10.3390/medicina59010113_

Round 1

Reviewer 1 Report

Dear authors, 

thank you for submitting the manuscript “Combined Humeral Head And Shaft Fractures: Outcome Following Intramedullary Nailing And Plating Combined Humeral Head and Shaft Fractures are not that often and always challenging for the surgeon. 

Some remarks:

The treated cases are quite old. In the last years new implants (especially nails) for osteoporotic fractures were developed. It would be interesting to investigate patients undergoing surgery the last few years. On the other hand it would be really informative for the reader to know exactly the implants you used. Please also describe the complications you observed. Was it always a displacement of the metaphysis or also of the head. It these cases screw tip cement augmentation could be useful.

Author Response

See the reply to reviewers in the attached file.

Reviewer 2 Report

Thanks for allowing me to review the manuscript entitled 

Combined Humeral Head And Shaft Fractures: Outcome Following Intramedullary Nailing And Plating”.

The authors have investigated the outcome of two different fixation methods of combined humeral head and shaft fractures. The retrospective study included 25 patients which were treated with nail (n=16) or plate (n=9) fixation. Outcome parameters included rates of complications, revision surgeries and functional outcome parameters (CMS).

However, even though fracture treatment of these fracture specimens have already been published before I personally overall think that this is a nicely performed piece of work with clinical relevance. Due to the number of included patients that were investigated, the follow up of at least 12 months including outcome parameters such as bony healing, complications and functional assessment, this study does add some new aspects to the current body of evidence and is worth to publish. Nevertheless, I have major concerns which need to be addressed before the manuscript could be published.

Point by Point recommendation:

1) Within the result-section of the abstract you mentioned n=4 reoperations in the nailing group, however, according to tables guess there were n=5 reoperations in this group.

2) In lines 24-26 of the abstract, you mentioned n=6 patients with secondary displacement of the metaphyseal fragments. In the result section of manuscript, you mentioned n=2 revisions according to displacement of the metaphyseal fragments, n=2 revisions after implant removal and n=1 loss of reduction. Please clarify. Have there been patients with relevant displacements without revision?

3) In the method section of the manuscript you define the outcome parameter “relevant secondary displacement”. However, within table 1 and the manuscript you also use the term “loss of reduction”. To enhance reader-friendliness I would recommend to present the results using one defined outcome parameter such as “relevant secondary displacement” or define the other terms before using it.

4) Table 1 features a patient with the implant type “retrograde”. Did you include a patient with retrograde nailing? Please add information. How many different surgeons performed the procedures? 

5) I would recommend to feature the outcome parameter bony healing. You mentioned n=1 revision due to a nonunion. Therefore, I assume that the other n=24 patients showed bony healing within the study period. However, because you mentioned n=2 patients with loss of reduction after implant removal, and n=1 patient with nonunion it might be interesting to know how you define bony healing. How long was the time interval between nailing and implant removal? You mentioned early removal of the nail? What was the reason for early implant removal?

6) In the method section of the manuscript, you mentioned the outcome-parameter ”implant failure”. However, the result-section does not feature the results of this outcome parameter. Have there been implant failures?

7) I would recommend to add some information according to the implant types; at least length of the nail and plates.

8) In lines 184-186 of the discussion section you stated that in cases of secondary metaphyseal displacement, the functional outcome was significantly worse regardless of the osteosynthesis hardware used (nail or plate). I still don’t get how many patients suffered secondary dislocation. Have there been secondary dislocations within the plating group? Maybe you can add this information to table one.

9) You concluded that the classic closed reduction and intramedullary nailing is not the best treatment option. But that comparable clinical outcome was obtained when comparing nailing with additional open cerclage or lag-screw fixation techniques versus plating with open reduction. According to what outcome parameters do you conclude that? According to your results within the ”classic nailing group” (n=7) n=2 revisions were obtained (2/7 = 28%); within the nailing group with additional cerlages or screws (n = 9) you observed n=3 revisions (3/9 = 33%). What significant differences did you observe between these two groups. Furthermore, within the plating group you found only one revision which was a nonunion (1/9=11%). I interpret these results as superior in comparison to nailing. 

Author Response

(The authors gave the same response as above.)

Reviewer 3 Report

This retrospective study enroll 25 patients with humeral head and shaft fracture undergoing humeral nail and long plating. Although the case number is small, this study provide ideas that plating provide less re-operation rate, less secondary displacement of the fragment and comparable functional outcomes in humeral head and shaft fracture than the nailing.

1.  Line 16 humeral head and shaft could be replaced by FHS for short

2.  Line 63 Figure 2 need to be optimized definately

3. Please make a table and summarize the classification used in the manuscript

4.  Please describe the characteristics of fracture treated by IM nail in more detail, and try to figure out the main reasons caused higher re-operation rate, such as differences in the extension of fracture split into shaft, poor reduction of the metaphyseal fragments, or lack of fixation of in metaphyseal area.

Author Response

(The authors gave the same response as above.)

Reviewer 4 Report

Dear authors,

I read your paper about associated ipsilateral humeral head and shaft fracture with great interest. Indeed, the presence of the so-called intermuscular fracture pattern makes the treatment challenging for any surgeon.

The study compares two different methods of internal fixation of these fractures, of course emphasizing the need to fix the intermuscular fracture.

The paper is well structured and concisely presented. Particularity, it contains a long follow-up period (up to 97 months).

Best regards!

Author Response

Thank you very much for your review.

Round 2

Reviewer 1 Report

Thank you for the revision of the manuscript. In my opinion a publication is possible.